# Computational Study of the Kinetics and Mechanisms of Gas-Phase Decomposition of *N*-Diacetamides Using Density Functional Theory

**DOI:** 10.3390/molecules29163833

**Published:** 2024-08-13

**Authors:** Oswaldo Luis Gabidia Torres, Marcos Loroño, Jose Luis Paz Rojas, Cecilio Julio Alberto Garrido Schaeffer, Thais Cleofe Linares Fuentes, Tania Cecilia Cordova Sintjago

**Affiliations:** 1Departamento Académico de Fisicoquímica, Facultad de Química e Ingeniería Química, Universidad Nacional Mayor de San Marcos, Lima 15081, Peru; 2Departamento Académico de Química Inorgánica, Facultad de Química e Ingeniería Química, Universidad Nacional Mayor de San Marcos, Lima 15081, Peru; jpazr@unmsm.edu.pe; 3Departamento Académico de Operaciones Unitarias, Facultad de Química e Ingeniería Química, Universidad Nacional Mayor de San Marcos, Lima 15081, Peru; cgarridos@unmsm.edu.pe; 4Departamento Académico de Química Orgánica, Facultad de Química e Ingeniería Química, Universidad Nacional Mayor de San Marcos, Lima 15081, Peru; tlinaresf@unmsm.edu.pe; 5Department of Natural Sciences, Santa Fe College, Gainesville, FL 32066, USA; tania.cordova-sintjago@sfcollege.edu

**Keywords:** DFT, diacetamides, pyrolysis, IBSI, NBO, reaction mechanism

## Abstract

In this research work, we examined the decomposition mechanisms of *N*-substituted diacetamides. We focused on the substituent effect on the nitrogen lone-pair electron delocalization, with electron-withdrawing and electron donor groups. DFT functionals used the following: B1LYP, B3PW91, CAMB3LYP, LC-BLYP, and X3LYP. Dispersion corrections (d3bj) with Becke–Johnson damping were applied when necessary to improve non-covalent interactions in the transition state. Pople basis sets with higher angular moments and def2-TZVP basis sets were also applied and were crucial for obtaining consistent thermodynamic parameters. The proposed mechanism involves a six-membered transition state with the extraction of an α hydrogen. Several conformers of *N*-diacetamides were used to account for the decrease in entropy in the transition state in the rate-determining state. All calculations, including natural bond orbital (NBO) analyses, were performed using the Gaussian16 computational package and its GaussView 6.0 visualizer, along with VMD and GNUPLOT software. The isosurfaces and IBSIs were calculated using MultiWFN and IGMPlot, respectively.

## 1. Introduction

Diacetamide and *N*-acetyl acetamide (CH_3_CONHCOCH_3_) exist as dimers in a crystalline state. The molecules are bound via hydrogen bonding C=O---HN, in which one carbonyl group of the molecule acts as a proton acceptor for the imide proton, while the other carbonyl group is not bound by hydrogen at all [1,2,3,4]. Due to this intermolecular hydrogen bonding, the geometry of diacetamides shows small differences in their structures when comparing the gas phase and X-ray electron diffraction geometries. The intramolecular hydrogen bonds C=O---HN cause an elongation of the C=O bond (0.01 Å) and a comparable shortening of the CN bond with respect to the values of the isolated molecule [5]. Figure 1 shows two models. In the first model (Figure 1a), an HN bond reproduces the cyclic dimer of the crystal. The second model (Figure 1b) describes a structure where two monomers are associated by a single C=O---HN hydrogen bond and form an open dimer structure. The latter model describes a disordered intermolecular interaction like that expected in a concentrated low-temperature matrix. Gas-phase electron diffraction also suggests that this diacetamide is a nearly planar molecule with a thermally averaged dihedral angle between the two acetyl groups of 36° [5,6].

It is generally accepted that the hydrolysis of acetamide is initiated by the formation of an *O*-protonated tautomer through the nucleophilic attack of a water molecule on carbonyl oxygen. However, the homogeneous gas-phase pyrolysis of acetamide produces ammonia (NH_3_), acetic acid (CH_3_COOH), and acetonitrile (CH_3_CN) as the main products. Alternate decarboxylation and dehydration reaction pathways were suggested, which dominate the unimolecular breakdown of analog acetic acid. Ammonia, acetic acid, and acetonitrile are also the main products of acetamide hydrolysis in supercritical water and during photodissociation [7]. One of the techniques used to study molecular decompositions is pyrolysis. The molecules are known to have limited thermal stability, which leads to the formation of smaller molecules, although the resulting fragments can also interact and generate larger compounds compared to the initial molecule. When the heating temperature is above 350 °C, chemical processes caused by thermal energy alone are called pyrolysis (Py). Chemical reactions caused by heat at lower temperatures (e.g., 250 °C) are called thermal decompositions [8,9,10,11]. Numerous pyrolysis products occur in nature; the toxicological and environmental implications of their presence are of considerable interest [12,13,14].

Figure 1 shows the proposed mechanism for the thermal decomposition of a *N*-(4-nitrophenyl) diacetamide given by AL-AWADI et al. [15], involving a six-membered ring, where the participation of the lone-paired electrons of nitrogen is very important. The kinetic data revealed that this reaction follows a first-order kinetics. The reactivity is modified when a *N*-aryl substituent is present, relative to a simple diacetamide.

Molecules with a delocalized p-system of electrons represent attractive targets for application in advanced functional materials. These molecules have junctions in which nitrogen actively participates. This work aims to provide more insight into the thermal decomposition of *N*-substituted diacetamides. The effects of substituents at the aromatic ring are discussed. To do so, the thermodynamic properties of diacetamides were computationally determined and then compared with the available experimental data.

## 2. Computational Methodology

The molecular structure of all compounds in this study, reactant, and products, were optimized at the level of theory used to calculate the thermodynamic parameters. The series includes N-substituted diacetamides, namely NX(COCH_3_)_2_ with X substituents: H, phenyl, or 4-nitrophenyl. These parameters were used to evaluate the kinetics of the decomposition reaction of N-substituted diacetamides and measure the effects of the electron donor and/or electron-withdrawing groups at the substituent on the nitrogen. Table 1 shows only the experimental activation energies obtained by AL-AWADI [15].

The Gaussian16 computational package revision C.01 WIN64-version 1.1 [16] was used in order to carry out all the structure optimizations studied with the following DFT functionals: B1LYP, B3PW91, CAMB3LYP, LC-BLYP, and X3LYP together with the basis sets 6-31+g(d,p), 6-31+(2d,2p) and 6-31+(3df,2p), 6-311+(2d,2p), 6311+(3df, 2p), and def2-TZVP [17,18]. All calculations were carried out to convergence, with no negative frequencies for the minimum energy structures, or a single negative frequency if a transition state is involved. To ensure that there was a correspondence between reactants and products, an intrinsic coordinate interaction (IRC) procedure was used, thus creating an energy interaction coordinate that connected the participating species through the activation energy. The topological properties of the electron density at the bonding critical point (bcp) of molecules were calculated according to Bader’s quantum theory of atoms in molecules (AIM). The reduced density gradient was obtained to determine the types of non-covalent interactions present in the structure. These calculations were carried out using the MultiWFN software [19]. To measure the strength of the bonds, the independent gradient model (IGM) was used [20,21,22,23,24,25]. Finally, frequency calculations allowed for obtaining zero-point vibrational energy (ZPVE), temperature corrections [E(T)], and absolute entropies [S(T)] assuming ideal gas behavior from the harmonic frequencies and moments of inertia by standard methods [26]. The Gibbs free energy change (ΔG^#^) between each reactant (R) and its transition state (TS), was estimated using the following equations:ΔG^#^ = ΔH^#^ − TΔS^#^(1)
ΔH^#^ = V^#^ + ΔZPVE + ΔE(T)(2)
where V^#^ is the potential energy barrier and ΔZPVE and ΔE(T) are the differences in ZPVE and temperature corrections between the TS and the R, respectively. Entropy values were estimated from a harmonic vibrational analysis.

## 3. Results and Discussion

The experimental results given by AL-AWADI entail a six-membered ring transition state, where an α hydrogen is extracted by the carbonyl oxygen; see Figure 1. According to Table 1, if X=H, the activation energy is 151.3 ± 2.7 kJ/mol; it rises significantly when hydrogen is substituted by a phenyl group or substituted up to an average energy of 190 kJ/mol. The effect is explained by a delocalization of electrons from nitrogen to the phenyl group. This energy difference is about 40 kJ/mol. This increase in energy can be explained computationally, making use of the density functional theory (DFT). For this purpose, several types of functionals—B3PW91, X3LYP, B1LYP, LC-BLYP, and CAM-B3LYP—and different Pople basis sets—with dispersion corrections and Becke–Johnson damping (d3bj)—were chosen. Additionally, an Ahlrichs def2-TZVP basis set was also used.

Three compounds were analyzed together with three possible mechanisms, NX(COCH3)_2_ X=H, phenyl, and p-nitro phenyl; see Figure 2. In the first mechanism, Figure 1 was adopted, as experimentally proposed by AL-AWADI (direct method). The second mechanism (Figure 3) involved the formation of dimers. Later, these were modeled in accordance with the formation of a transition state according to the final products found experimentally. The third mechanism involved an initial four-membered cyclic transition state and an intermediate and six-membered cyclic transition state leading to product formation; Figure 4.

In dimers, hydrogens are transferred from both molecules and not from one in an intramolecular form. Calculations were performed at the X3LYP/def2TZVP level. The choice of this model will be seen later when we attack the direct mechanism of diacetamide breakdown. For the case of X=H, the calculation shows a dimer with an oscillation frequency of −1459.79 cm^−1^, typical of a hydrogen vibration, with an enthalpy of −723.385921 Ha, which, with respect to the minimum energy monomer −361.731421 Ha, would yield an activation enthalpy of 201.96 kJ/mol, which is quite high. It was just as high when the hydrogen atom was replaced by a more effective electron attractor such as chlorine—that is, X=Cl—and the same functional X3LYP. In this case, the enthalpy of the monomer would be −821.262913 Ha; compared to −1642.441213 Ha for the formation of the transition state; this would imply spending 222.14 kJ/mol of energy, with an oscillation frequency of −1475.01 cm^−1^. A similar result emerged when the phenyl group, X=phenyl, was used, obtaining an energy of 212.16 kJ/mol to form. The high calculated energy values led us not to use this model in the subsequent calculations.

In the case of the second model or mechanism, in Figure 4, transition state 1 is modeled by placing two fragments perpendicularly. This would generate an intermediate state (A) prior to the formation of the products, that is, a cetene and an amide. The intermediate compound would involve a transitional state of a seven-member ring, as enclosed below, when products are formed.

Table 2 shows the computational results for the mechanism proposed for Figure 4, with five functionals and four basis sets with different polarization functions in addition to the basis set of Ahlrichs def2-TZVP being used. The calculated thermodynamic parameters were consistent with the values found experimentally, even with signs of entropy. Among all, the functional B3PW91 showed a better agreement with the activation energy values.

At this point, it was decided to apply the direct mechanism for the case of NX(COCH_3_)_2_ X=H. Table 3 shows the calculation results for diacetamide (X=H), a basis set of Def2-TZVP. As can be seen, it was not possible to reconcile the results according to the experimentally obtained activation energy value of about 150 kJ/mol. However, the proposed mechanism was adequate when hydrogen was replaced by a chlorine or methyl atom. In the first case, it is possible that chlorine will donate electron density to the nitrogen, strengthening the N-Cl bond, which can help polarize the C=O and increase the attractive force of methyl alpha hydrogens. The calculated thermodynamic parameters were consistent with the values found experimentally, even with signs of entropy. In the case of methyl, there is an electron donor effect. The excellent results found for the functional X3LYP and basis set of def2-TZVP, highlighted in black in Table 3, led us to use this model in the case of dimers.

For the N-phenyl diacetamide (X=phenyl), our computational calculations show that the phenyl group is oriented perpendicular to the plane C(O)NC(O). This conformation prevents electron delocalization with the aromatic ring; regardless of whether the phenyl group is substituted with electron-withdrawing or electron donor groups, the experimental activation energy (Table 1) is approximately 30% of the value compared to hydrogen. In order to explain these values and demonstrate that nitrogen cannot share its electron pair with the phenyl ring, we proposed different actions. In the first place, we found three energetically probable conformers, with three possible methyl orientations (Figure 2); once the most stable structure was obtained (Figure 2C), the phenyl was rotated in 10° intervals from 90 to 180°. We calculated its charge, intrinsic bond strength index (IBSI), and energy change when the phenyl orientation was modified from perpendicular to in-plane orientations. The results are shown in Table 4.

Any attempt to place the aromatic ring in the same plane as C(O)NC(O) was rendered impossible. To calculate the energy cost, a redundant coordinate calculation was made in which the torsional barrier (TB) was determined. These calculations were performed at the B3PW91/6-311+G(3df,2p) level of theory.

To determine the torsional barrier (TB), the conformation scans started with the phenyl perpendicular ring to the plane C(O)NC(O), shown in structure A with an angle of 90.06° and energy of −592.92 Ha, and ended with the phenyl ring rotated at 180° with respect to the initial conformation. In the final conformation, shown in structure B, the phenyl ring is coplanar to the plane C(O)NC(O) and the energy is −592.90 Ha. The energy change difference is 39.96 kJ/mol. The result is shown in Figure 3, along with the total energy scanned. The atom numbers in structure A of Figure 3 depict the atom numbering for the NBO calculation. The IBSI parameters are shown in Table 4.

Figure 4 now shows the potential energy surface resulting from the rotation of two dihedral scan coordinates (SC1: C2-N1-C14-C15 and SC2: C14-N1-C2-C3; see Figure 3A for the atom labels) and the results from the additional movement of the groups of atoms attached to the nitrogen atom. The figure shows two black arrows; with the top one, the transition state is observed, and the one below corresponds to the global minimum found. When taking the difference in energies between these points (591.533275–591.461992) Ha, it corresponds to 187.15 kJ/mol, which is very close to the value measured experimentally. The IBSIs, in Table 4, show that when the benzene ring was coplanar to the C(O)NC(O) plane, electron density shifted from the aromatic ring towards the carbonyl carbon. That effect is not appreciable when the ring is perpendicular to the C(O)NC(O) plane. A slight displacement of the NBO charges by the ring can also be seen at the N1-C14 bond of contribution when the system is in the C(O)NC(O) plane. Table 5 shows the thermodynamic parameters using the functional B3PW91. The best agreement was found when X=hydrogen. Close attention was paid to two variables: the sign of entropy and the value of the activation energy.

As can be seen in Table 5, the Pople basis set 6-31G and functional B3Pw91 generate the value of 186.95 kJ/mol, which is very close to the experimental one. However, it is observed that the entropy has a positive value of 17.61 J/K.mol. When we improve the basis sets and add a polarization function (d), the entropy changes to negative and the activation energy drops to 178.07 kJ/mol. Nonetheless, considering the experimental error, this was a better value. The entropy value decreases as the basis sets improve. When using the B1LYP functional, we found something interesting; the results converge very well, but they worsen as the basis set size is increased. This induced us to investigate whether there was a basis set superposition error (BSSE) effect [27] since in the transition state, two molecules are separated. This basis effect was calculated to avoid the underestimation of the results. From Table 5, it is clear that BBSE correction did not result in a significant change (only about 6 kJ/mol); consequently, we did not continue using it.

In the case of 4-nitrophenyl, in Figure 5, as has been shown for the case of the phenyl group when the ring is perpendicular to the plane of the C(O)NC(O), the electron-withdrawing or electron donor group has no appreciable effect on the reaction rate. In fact, when phenyl is used, the value of the experimentally calculated activation energy is 185.7 (± 7.5) kJ/mol, and compared to 4-nitrophenyl, this value is 192 kJ/mol, which is the same as within the experimental error. To prove this assertion, we calculated the activation energy at the B3PW91 level of theory (including a dispersion correction with Becke–Johnson damping (d3bJ) and the 6-311+G(3df,2p) basis set).

The calculation of the thermodynamic properties at the B3PW91-D3BJ/6-311+G(3df,2p) level of theory yields an activation energy value of 150.69 kJ/mol, a value that is comparable to that calculated for the case of phenyl at 156.52 kJ/mol. Regarding the entropy change of −8.64 J/K mole, its value was consistent with a negative value. All the above results allow us to infer that the reaction mechanism is the same as for *N*-phenyl, a direct mechanism. An even better result is obtained using the functional LC-BLYP and the def2-TZVP basis set.

Table 6 shows the thermodynamic results maintaining the base set def2-TZVP and varying the type of functional. A correspondence between the calculated and experimental energy of activation was observed when using the LC-BLYP/def2-TZVP level of theory; that is, this functional provides the best description of our system and shows that it does not matter in which position the nitro group is located to characterize the displacement of the electrons towards the aromatic ring.

In addition to the case of the phenyl, the case of 4-nitrophenyl offered an opportunity to explore some of the features of the natural bond analysis (NBO) for studying both inter- and intramolecular interactions, as well as interactions between bonds, charge transfers, or conjugative interactions in molecular systems. In fact, the NBO method gives very useful information about the interactions between full and empty orbitals, which is equally important in the reactivity of interacting systems. This last feature is accomplished by considering all possible interactions between natural bond orbitals (NBOs), donors, and acceptors and then estimating their involved energies using second-order perturbation theory [28]. For each donor orbital (i), there is an acceptor orbital (j), and the associated stabilization energy E(2), due to the electronic delocalization, is given by
(3)$$E\left(2\right)=∆E_{i,j}=\frac{q^{i}F(i,j)}{\left(E_{j}-E_{i}\right)}$$
where $q^{i}$ is the occupation of the orbital, i.e., the number of electrons, $E_{j}$ and $E_{i}$ are the energies involved and F(i,j) are the elements outside the diagonal of the Fock matrix. In this sense, a great value of E(2) is indicative of the intensity of the interaction between the electron-donor and electron-acceptor species, meaning that there is a greater tendency for electrons to be donated and, therefore, a greater conjugation of the studied system.

For our case, natural bond orbital analysis was performed using the B3PW91/6-311+G(3df,2p) d3bj method. For the specific case of the 4-nitrophenyl diacetamide, shown in Figure 5, several points of the IRC graph of Appendix A were studied, passing the structure from a reactant to the products. Here, we selected the bonds N1-C14, N1-C7, C7-O13, and C21-N24. Such bonds would show the path of the electrons or the conjugation that could be present. Appendix A is formed by 34 points before the transition state and 40 points from then, up to the formation of the products. There, we selected some optimized structures and constructed Table 7. It shows the strong interaction of N1 electrons towards the C7-O13 bond; this electron migration favors the extraction of hydrogen β. During this process, it is also observed that before reaching the transition state, no involvement of the lone electron pairs of nitrogen is observed. After the transition state, the IBSI value for the N1-C2 bond is not displayed, as it is less than 0.05. Within the nitro group, there is a strong conjugation above 170 a.u. for the value of the perturbation energy. A total of eight structures or points were studied, including the transition state. IBSI values for the N1-C14 and N24-C21 bonds remained constant during the mechanistic process. These results reinforce the non-conjugate participation of the benzene ring, and, therefore, the reaction rate would not be affected by the presence of the nitro group in the phenyl. The NBO was included in the Gaussian16 package and the IBSIs were calculated using the IGMPLOT program.

Bond orders can also be used as descriptors of covalent bonding, involving two atoms, and they are used very widely in chemistry. Indeed, notions of single, double, and triple bonds (with bond orders of 1, 2, and 3, respectively) are very common. In this work, attention was paid to the binding between the groups (X=H, phenyl, or Cl) and nitrogen, in search of any modification of the electron density. In particular, the Mayer bond order is a natural extension of the Wiberg bond order, which has been proven extremely useful in bonding analysis using semi-empirical computational methods and Mulliken population analysis to develop ab initio theories. Figure 6 shows a Mayer bond order analysis for the case of NX(COCH_3_)_2_ X=H, phenyl, and Cl. Most details can be visualized in the Appendix A. The biggest contributor to the bond strength is the Chloro group followed by the phenyl group. In the case of X, when it is replaced by hydrogen, there is not much of a change noticed—only during the transition state.

## 4. Conclusions

The calculations carried out during this work allowed us to identify and study different types of reaction mechanisms from the one proposed in the experimental study by AL-AWADI involving a six-membered ring. In that proposal, the participation of the lone electron pair of nitrogen played an important role, assuming delocalization effects by aromatic substituents at the nitrogen atom. However, it was not possible to use the same mechanism for all molecules studied. Several mechanisms were suggested depending on the substituents present. When at NX(COCH_3_)_2_X, we substituted X=H; the calculated activation energies suggested a two-stage mechanism, different from the case of X=phenyl, where the best agreement between experimental and calculated parameters led to one direct mechanism at which the phenyl was almost perpendicular to the C(O)NC(O) plane. A small rotational energy barrier of 39.96 kJ/mol was sufficient to prevent the electron delocalization between the nitrogen and the aromatic ring, thereby explaining the negligible effect of the ring substituent in the reaction rates.

## Data Availability

Any data not presented in the link provided in the supplementary materials section, can be available on request from the corresponding authors.

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
