# Peer review of "Computational Study of the Kinetics and Mechanisms of Gas-Phase Decomposition of N-Diacetamides Using Density Functional Theory"

_molecules, 2024, doi:10.3390/molecules29163833_

Round 1

Reviewer 1 Report

Comments and Suggestions for Authors

The paper "Computational study of the kinetics and mechanism of gas-phase decomposition of N-diacetamides using density functional theory" by Oswaldo Gabidia T. et al., contains some interesting results and may be suitable for publication in Molecules. However, the current version of the manuscript is not acceptable for publication. The authors should address the comments listed below. Also, in general the quality of the text is not good, there are many grammatical mistakes.  

Major comments:

1) The authors should explain the importance of the particular studied compounds, otherwise the importance of the work is unclear to a non-specialist.

2) The BSSE correction was applied sporadically, but it should be mentioned in computational details and applied more systematically. It should be clear for which systems it was applied or not and why. 

3) "different Pople basis sets, with dispersion functions (gd3bj)" - strange wording. It should be clear which dispersion CORRECTION or functional the authors used. I do not understand what they mean by "dispersion functions".

4) The authors should define and give formulas that they have used to calculate Delta G and Delta S.

5) There are significant differences between results obtained with different functionals, but the physics underlying these differences is not discussed. It is also not explained why these particular functionals were chosen. I think the authors should clarify their choices and explain physics behind at least some of the most prominent differences between functionals.

Minor comments:

"...it rises significantly when hydrogen is substituted by a phenyl group or substituted." - unfinished sentence?

"making use of the functional density theory (DFT)" - density functional theory

Table 4: make clear the meaning of "tipo"; also, what is "grades"? did you mean degrees?

Figure 3 - what are the units of the number on the plot (39.96)?

Figure 4 - explain to the readres what SC1 and SC2 mean

"the use of the LC-BLYP/Def2-TZVP level of theory was adequate to describe our system" - clarify this sentence; what does "adequate" mean in this case? how this was determined?

Comments on the Quality of English Language

There are many punctuation errors, some minor English grammar problems.

Author Response

Reviewer #1:

The paper "Computational study of the kinetics and mechanism of gas-phase decomposition of N-diacetamides using density functional theory" by Oswaldo Gabidia T. et al., contains some interesting results and may be suitable for publication in Molecules. However, the current version of the manuscript is not acceptable for publication. The authors should address the comments listed below. Also, in general the quality of the text is not good, there are many grammatical mistakes.
Our answer:
Thank you, I really appreciate your feedback we have done our best to correct all possible grammatical errors. Please, allow me to continue with his concerns.
Reviewer
1) The authors should explain the importance of the particular studied
compounds, otherwise the importance of the work is unclear to a nonspecialist.
Our answer:
We have now introduced a small paragraph to treat your concern
Reviewer
2) The BSSE correction was applied sporadically, but it should be mentioned in
computational details and applied more systematically. It should be clear for
which systems it was applied or not and why.
Our answer:
Thanks, we only used the BSSE correction to calculate the effect on the activation
energy, but when we realized that its effect was no more than 6 kJ/mol and not a
major change we did not continue to use it.
Reviewer
3) "different Pople basis sets, with dispersion functions (gd3bj)" - strange
wording. It should be clear which dispersion CORRECTION or functional the
authors used. I do not understand what they mean by "dispersion functions".
Our answer:
Thanks, we have now used the correct symbology in the manuscript.
Reviewer
4) The authors should define and give formulas that they have used to calculate
Delta G and Delta S.
Our answer:
The values of both delta G and S were extracted directly from Gaussian16's results.
However, the activation energy values were calculated from enthalpy by adding RT,
as indicated by the activated complex theory.
Reviewer
5) There are significant differences between results obtained with different
functionals, but the physics underlying these differences is not discussed. It
is also not explained why these particular functionals were chosen. I think the
authors should clarify their choices and explain physics behind at least some
of the most prominent differences between functionals.
Our answer:
Our study was limited only to the use of DFT functionals, and several of them were
used in agreement with other computational calculations involving nitrogen. We
understand that the DFT functional is made up of a complex contribution of
correlational and interchange parts, such as series involving constants. The values
of the constants determine the type of functional or contribution. Long range
corrections (LC-) in combination with B (exchange), LYP (correlation) parts resulted
good for our calculation in the case of X=phenyl, but it is not always the case.
Minor comments:
Reviewer:
"...it rises significantly when hydrogen is substituted by a phenyl group or substituted." - unfinished
sentence?
Our answer:
Yes, thanks, the sentence was completed in the manuscript.
Reviewer:
"making use of the functional density theory (DFT)" - density functional theory
Our answer:
Thanks, it was fixed.
Reviewer:
Table 4: make clear the meaning of "tipo"; also, what is "grades"? did you mean degrees?
Our answer:
Thanks. It was fixed.
Reviewer:
Figure 3 - what are the units of the number on the plot (39.96)?
Our answer:
Yes, we have fixed, we include kJ/mol.
Reviewer:
Figure 4 - explain to the readres what SC1 and SC2 mean
Our answer:
A short explanation was included. Thanks.
Reviewer:
"the use of the LC-BLYP/Def2-TZVP level of theory was adequate to describe our system" - clarify
this sentence; what does "adequate" mean in this case? how this was determined?
Our answer:
Yes, you are right, we re-write the sentence to be more precise.
I thank you for your time dedicated to this manuscript, I really appreciate it! Please,
notices that all the changes you have made are highlighted in yellow in the
manuscript called “marked main manuscript”.

Sincerely,

            Prof. Marcos A. Loroño G                                        

Reviewer 2 Report

Comments and Suggestions for Authors

See attchec PDF.

Author Response

Reviewer #2:

The content should be of interest to both theoretical and experimental researchers. We recommend that the manuscript be modified to address the following questions and comments.
Our answer:
Thanks for your comments. Reviewer
1) The graphical abstract figure is not discussed at all in the manuscript; it is an Attractive figure but seems to be drawn only to attract the reader's interest. I recommend the authors to discuss on this figure in the graphical abstract.
Our answer:
Thanks, the recommendation was considered.
Reviewer
2) The authors propose reaction pathways that differ from the experimental one.
However, when they evaluate the quality of DFT-functionals and Basis Sets,
they assume the reaction pathway proposed by the experimenters. Is this not
a contradiction? In other words, the authors' position is that the reaction
pathways giving the experimental values in Table 1 should be unknown.
Our answer:
At first, we wanted to follow the same reaction mechanism proposed
experimentally, during the experiment we realized that it was simply not possible,
but only in the case when X = hydrogen. We believe that the rest of the molecules
follow the experimentally proposed mechanism, with the only exception that the
migration of the pair of electrons from the nitrogen is determined by a delicate
mechanism implied in the text, when the aromatic ring is twisted. We are sure it may
be subject to further studies.
Reviewer
3) Page7, line 207, At this point it was decided to apply the direct mechanism
for the case, NX(COCH3)2 X=H. Explain what the reason is.
Our answer:
Thanks, I have already answered for the question 2.
Reviewer
4) In the discussion of Figure 2, why did the author make the Basis Set worse?
Why not use the Basis Set and functional verified in Table 2?
Our answer:
The mechanism of Scheme IV was an alternative and explains the experimental
results very well, but unfortunately, they were not applied to the rest of the
molecules, since it involved the movement of an entire aromatic ring.
Reviewer
5) Page 11, line 309, the argument that the results agree with the experimental
values by making the basis function worse is nonsense and unacceptable.
Our answer:
I do respect your opinion but after many calculations and possibilities, we got
these results.
Reviewer
6) Page 11, line 331, what does better result mean? Again, why change the Basis
Set at the analysis stage? I can only think that they are choosing a basis
function that gives a convenient result.
Our answer:
We appreciate your concern, but it is not the result of testing functionals for the sake
of testing, we kept a horizon in mind, testing traditional functionals, and those
involving non-covalent interactions with different basis sets of the Pople type with
those of Ahlrichs def2-TZVP.
Minor comments:
Reviewer:
1) The caption of Figure 1 and the explanation in the manuscript are confusing.
Our answer:
Yes, thanks, the caption was improved, and we have also changed the Figure itself.
Reviewer:
2) In Figure 1, (a) and (b) are explained, but (b) is not present.
Our answer:
Thanks, it was fixed.
Reviewer:
3) Table 4 is on page 9, but the definitions of N1, C2, and C14 are in Figure 5 on page 14. They
are too far apart. I had difficulty finding this definition; Specify where the definitions of the
atom-numbers on Table 4 are located.
Our answer:
Yes, the symbology used for Table 4 is connected to the numbering of the symbols
in Figure 3. Thanks for the detail.
Reviewer:
4) Figure 3 does not have the definition of angle; it should be shown in the graphs of A and B.
Also, the left and right sides are opposite in A and B. This is extremely difficult to
understand.
Our answer:
Thanks for the spot. We now include a short sentence in the caption.
Reviewer:
5) Reference number should be written in a consistent manner. Example, [15] or (15).
Our answer:
Yes, thanks. We have now changed in a consistent manner.
Reviewer:
6) ”functional density” should be “density functional”.
Our answer:
Yes, we fixed.
Reviewer:
7) in several places, Scheme 4 ---> Scheme IV.
Our answer:
Thanks. We have now fixed (Scheme IV).
I thank you for your time dedicated to this manuscript, I really appreciate it! Please,
notices that all the changes you have made are highlighted in green in the
manuscript called “marked main manuscript”.

Sincerely,  

            Prof. Marcos A. Loroño G                                        

Reviewer 3 Report

Comments and Suggestions for Authors

The manuscript studied the reaction mechanisms and dynamics of N-substituted diacetamides using density functional theory(DFT). It indicated that the lone pair electron delocalization of nitrogen is important in the transition of the experiment by AL-AWADI. The reaction mechanism of NX(COCH3)X with different substituents are different. The rotation energy barrier is high enough to prevent the electron delocalization and it leads to the difference of the reaction rate with different ring substituent.  

Here are my comments: 

1. In Page 9 figure2, the energy unit should be 'Ha' not 'ha'.  What is the energy resolution of your calculation? Does it necessary to include 6 digit? (E: -592.407279.)  

2. I suggest change all the energy in Hartree to a standard unit, like eV or kJ/mol. 

3. In Table 7, why the first column is 90 degree rotated? 

4. Why there are so many blank in Table 7? And the font in Table 7 is not consistent. 

5. I suggest to change Page4 line115 "GAUSSIAN16" to "Gaussian16" to be consistent to the manuscript. 

6. A small typo. Page 16 Line 455 "deslocalization" 

Author Response

Reviewer #3:

The manuscript studied the reaction mechanisms and dynamics of N-substituted diacetamides using density functional theory(DFT). It indicated that the lone pair electron delocalization of nitrogen is important in the transition of the experiment by AL-AWADI. The reaction mechanism of NX(COCH3)X with different substituents are different. The rotation energy barrier is high enough to prevent the electron delocalization and it leads to the difference of the reaction rate with different ring substituent.
Our answer: Thank you, you are right, the lone pair electro delocalization is very important. In the case of the Phenyl Group, it is shown that there is a barrier that prevents the connection with the aromatic ring, here, only a small twist is enough for the systems to connect. When X=H is replaced, it was more difficult to model, so an alternative mechanism was proposed and it was evident when chlorine was present, it was clearly seen that it contributes to accelerate the reaction by donating its electrons, where hydrogen could not.
Reviewer
1. In Page 9 figure2, the energy unit should be 'Ha' not 'ha'. What is the energy resolution of your calculation? Does it necessary to include 6 digit? (E: -592.407279.)
Our answer: Yes, I have changed ha to Ha, thanks! About the resolution, I consider that, by leaving more digits, I emphasize that the changes were very small.
Reviewer
2. I suggest change all the energy in Hartree to a standard unit, like eV or kJ/mol.
Our answer: Thanks for your suggestion, but for now we consider it is best to leave Hartree, since it is cohesive with the rest of the paper.
Reviewer
3. In Table 7, why the first column is 90 degree rotated?
Our answer: It was an aesthetic choice, but we have changed back to zero degrees.
Reviewer
4. Why there are so many blank in Table 7? And the font in Table 7 is not consistent.
Our answer:
Ah, right. The spaces were done intentionally. Since not all bonds were involved in the electron-NBO donor-acceptor pair (for example, N1 only involved points 1 to 4, while C7 to the transition state and points 6 to 8), we were looking for the best way to show the relationships formed between the donor-acceptor pair and the IBSI indexes. The font has been corrected, thank you for the observation.
Reviewer
5. I suggest to change Page4 line115 "GAUSSIAN16" to "Gaussian16" to be consistent to the manuscript.
Our answer:
Yes, thanks, we made the changes accordingly. Reviewer
6. A small typo. Page 16 Line 455 "deslocalization"
Our answer:
It was fixed, thank you!
As a final comment, I thank you for all the feedback, I appreciate your concern and time, dedicated to improving the manuscript, to make it. Please, notices that all the changes you have made are highlighted in red in the manuscript called “marked main manuscript”.

Sincerely,

Prof. Marcos A. Loroño G                                        

Round 2

Reviewer 1 Report

Comments and Suggestions for Authors

I am satisfied with the answers to my concerns, except for comments 4:

4) The authors should define and give formulas that they have used to calculate Delta G and Delta S

The authors replied that they took the numbers from Gaussian output. However, this targets only those who use Gaussian and know how these quantities are calculated. To improve readability for a wider audience, I recommend the authors to include the formulas in the text, explaining different contributions (or cite a reference where these formulas are clearly written), and that vibrational contributions were calculated in harmonic approximmation (I guess).

Author Response

Reviewer

  • “The authors should define and give formulas that they have used to calculate Delta G and Delta S”. The authors replied that they took the numbers from Gaussian output. However, this targets only those who use Gaussian and know how these quantities are calculated. To improve readability for a wider audience, I recommend the authors to include the formulas in the text, explaining different contributions (or cite a reference where these formulas are clearly written), and that vibrational contributions were calculated in harmonic approximmation (I guess).

 Our answer:

My apologies, we have now introduced a paragraph explaining how we calculated the Delta G and Delta S, and the formulas, and I have also added a new reference, the book of Dr. McQuarrie.

Reviewer 2 Report

Comments and Suggestions for Authors

See the attached PDF.

If the manuscript has been revised, do not just revise it, but indicate in the "Author Response" a summary of the changed sentences or changed points. This treatment will make a much better impression on the REVIEWER.

Author Response

Reviewer

  • The graphical abstract figure is not discussed at all in the manuscript; it is an Attractive figure but seems to be drawn only to attract the reader's interest. I recommend the authors to discuss on this figure in the graphical abstract.

Our answer:

Thanks. We have now written a short explanation for the NCI plots. However, regarding your concern, we did calculate the NCI plot for all of the structures, but they were not included in this manuscript, in the end, the one shown was left only to attract the reader's interest.  We detected non-covalent interactions using NCI plots and the IBSI calculations afterward. I agreed they looked great, but our interest was the IBSI calculations.

Reviewer

  • Shouldn't all reaction pathways be tried for any substituent of X= H, Phenyl, Cl? It does not seem a reasonable way to reach a conclusion.

Our answer:

My apologies. I did not follow you, correctly. Please, let me explain it. We did try only one mechanism for all the structures. However, when X= hydrogen the calculated energy was too high, no matter which basis sets or functional used. In the conclusion section, it is emphasized that the substituent bound to nitrogen is important for it to share its electrons. An aromatic system like benzene was important, but unfortunately, it is out of the plane, however, a slight twist could connect orbitals and facilitate electron sharing.

Reviewer                                                                                              

  • In the discussion of Figure 2, why did the author make the Basis Set worse? Why not use the Basis Set and functional verified in Table 2?

 Our answer:

Among all the functional B3PW91 showed a better agreement with experimental-theoretical values, several functionals were tried and basis sets of People and Aldrich in search of the best solution. 

Reviewer:

  • Table 4 is on page 9, but the definitions of N1, C2, and C14 are in Figure 5 on page 14. They are too far apart. I had difficulty finding this definition; Specify where the definitions of the atom-numbers on Table 4 are located.

 Our answer:

No please, the symbology used for Table 4 is connected to the numbering of the symbols in Figure 3, where we have now defined the scan coordinates to make it more understandable.

Reviewer:

  • The definition of “Scan Coordinate” is not given. In other words, Scan Coordinate is undefined. What molecular structure does the zero in Scan Coordinate refer to, and what structure does the 180 in Scan Coordinate refer to?

Our answer:

To obtain the data in Table 4 and the graph in Figure 4, two independent dihedral coordinates were selected, one that rotates the ring and the other taking care of the acetamides. With this, the structure of lower energy was determined and used to calculate the activation energy.

Reviewer:

 The authors do not answer the second comment. “Also, the left and right sides are opposite in A and B”. In other words, in A, phenyl is attached to the right-hand side, whereas in B, phyenyl is attached to the left-hand side. Was this reversed out of necessity? This reversal confuses the readers.

Our answer:

Yes, you are right. The figure has been modified accordingly.

Reviewer 3 Report

Comments and Suggestions for Authors

The revised manuscript and author's response effectively addressed all the questions. 

Author Response

Reviewer #3:

The revised manuscript and author's response effectively addressed all the questions. 

Our answer:

Thank you for all the feedback, I appreciate your time.
